# NASOA: Towards Faster Task-oriented Online Fine-tuning

## Abstract

Fine-tuning from pre-trained ImageNet models has been a simple, effective, and popular approach for various computer vision tasks. The common practice of fine-tuning is to adopt a default hyperparameter setting with a fixed pre-trained model, while both of them are not optimized for specific tasks and time constraints. Moreover, in cloud computing or GPU clusters where the tasks arrive sequentially in a stream, faster online fine-tuning is a more desired and realistic strategy for saving money, energy consumption, and CO2 emission. In this paper, we propose a joint Neural Architecture Search and Online Adaption framework named NASOA towards a faster task-oriented fine-tuning upon the request of users. Specifically, NASOA first adopts an offline NAS to identify a group of training-efficient networks to form a pretrained model zoo. We propose a novel joint block and macro level search space to enable a flexible and efficient search. Then, by estimating fine-tuning performance via an adaptive model by accumulating experience from the past tasks, an online schedule generator is proposed to pick up the most suitable model and generate a personalized training regime with respect to each desired task in a one-shot fashion. The resulting model zoo[1] is more training efficient than SOTA NAS models, e.g. 6x faster than RegNetY-16GF, and 1.7x faster than EfficientNetB3. Experiments on multiple datasets also show that NASOA achieves much better fine-tuning results, i.e. improving around 2.1% accuracy than the best performance in RegNet series under various time constraints and tasks; 40x faster compared to the BOHB method.

## 1 Introduction

Fine-tuning using pre-trained models becomes the de-facto standard in the field of computer vision because of its impressive results on various downstream tasks such as fine-grained image classification (Nilsback & Zisserman, 2008; Welinder et al., 2010), object detection (He et al., 2019; Jiang et al., 2018; Xu et al., 2019) and segmentation (Chen et al., 2017; Liu et al., 2019). Kornblith et al. (2019); He et al. (2019) verified that fine-tuning pre-trained networks outperform training from scratch. It can further help to avoid over-fitting (Cui et al., 2018) as well as reduce training time significantly (He et al., 2019). Due to those merits, many cloud computing and AutoML pipelines provide fine-tuning services for an online stream of upcoming users with new data, different tasks and time limits. In order to save the user's time, money, energy consumption, or even CO2 emission, an efficient online automated fine-tuning framework is practically useful and in great demand. Thus, in this work, we propose to explore the problem of faster online fine-tuning.

The conventional practice of fine-tuning is to adopt a set of predefined hyperparameters for training a predefined model (Li et al., 2020). It has three drawbacks in the current online setting: 1) The design of the backbone model is not optimized for the upcoming fine-tuning task and the selection of the backbone model is not data-specific. 2) A default setting of hyperparameters may not be optimal across tasks and the training settings may not meet the time constraints provided by users. 3) With the incoming tasks, the regular diagram is not suitable for this online setting since it cannot memorize and accumulate experience from the past fine-tuning tasks. Thus, we propose to decouple our faster fine-tuning problem into two parts: finding efficient fine-tuning networks and generating optimal fine-tuning schedules pertinent to specific time constraints in an online learning fashion.

---

[1]The efficient training model zoo (ET-NAS) has been released at: `https://github.com/NAS-OA/NASOA`

Recently, Neural Architecture Search (NAS) algorithms demonstrate promising results on discovering top-accuracy architectures, which surpass the performance of hand-crafted networks and saves human's efforts (Zoph et al., 2018; Liu et al., 2018a;b; Radosavovic et al., 2019; Tan et al., 2019b; Real et al., 2019a; Tan & Le, 2019; Yao et al., 2020). However, those NAS works usually focus on inference time/FLOPS optimization and their search space is not flexible enough which cannot guarantee the optimality for fast fine-tuning. In contrast, we resort to developing a NAS scheme with a novel flexible search space for fast fine-tuning. On the other hand, hyperparameter optimization (HPO) methods such as grid search (Bergstra & Bengio, 2012), Bayesian optimization (BO) (Strubell et al., 2019a; Mendoza et al., 2016), and BOHB (Falkner et al., 2018) are used in deep learning and achieve good performance. However, those search-based methods are computationally expensive and require iterative "trial and error", which violate our goal for faster adaptation time.

In this work, we propose a novel Neural Architecture Search and Online Adaption framework named NASOA. First, we conduct an offline NAS for generating an efficient fine-tuning model zoo. We design a novel block-level and macro-structure search space to allow a flexible choice of the networks. Once the efficient training model zoo is created offline NAS by Pareto optimal models, the online user can enjoy the benefit of those efficient training networks without any marginal cost. We then propose an online learning algorithm with an adaptive predictor to modeling the relation between different hyperparameter, model, dataset meta-info and the final fine-tuning performance. The final training schedule is generated directly from selecting the fine-tuning regime with the best predicted performance. Benefiting from the experience accumulation via online learning, the diversity of the data and the increasing results can further continuously improve our regime generator. Our method behaves in a one-shot fashion and doesn't involve additional searching cost as HPO, endowing the capability of providing various training regimes under different time constraints.

Extensive experiments are conducted on multiple widely used fine-tuning datasets. The searched model zoo ET-NAS is more training efficient than SOTA ImageNet models, e.g. 5x training faster than RegNetY-16GF, and 1.7x faster than EfficientNetB3. Moreover, by using the whole NASOA, our online algorithm achieves superior fine-tuning results in terms of both accuracy and fine-tuning speed, i.e. improving around 2.1% accuracy than the best performance in RegNet series under various tasks; saving 40x computational cost comparing to the BOHB method.

In summary, our contributions are summarized as follows:

- To the best of our knowledge, we make the first effort to propose a faster fine-tuning pipeline that seamlessly combines the training-efficient NAS and online adaption algorithm. Our NASOA can effectively generate a personalized fine-tuning schedule of each desired task via an adaptive model for accumulating experience from the past tasks.

- The proposed novel joint block/macro level search space enables a flexible and efficient search. The resulting model zoo ET-NAS is more training efficient than very strong ImageNet SOTA models e.g. EfficientNet, RegNet. All the ET-NAS models have been released to help the community skipping the computation-heavy NAS stage and directly enjoy the benefit of NASOA.

- The whole NASOA pipeline achieves much better fine-tuning results in terms of both accuracy and fine-tuning efficiency than current fine-tuning best practice and HPO method,e.g. BOHB.

## 2 RELATED WORK

**Neural Architecture Search (NAS).** The goal of NAS is to automatically optimize network architecture and release human effort from this handcraft network architecture engineering. Most previous works (Liu et al., 2018b; Cai et al., 2019b; Liu et al., 2018a; Tan et al., 2019a; Xie et al., 2019; Howard et al., 2019) aim at searching for CNN architectures with better inference and fewer FLOPS. Baker et al. (2017); Cai et al. (2018); Zhong et al. (2018) apply reinforcement learning to train an RNN controller to generate a cell architecture. Liu et al. (2018b); Xie et al. (2019); Cai et al. (2019b) try to search a cell structure by weight-sharing and differentiable optimization. Tan & Le (2019) use a grid search for an efficient network by altering the depth/width of the network with a fixed block structure. On the contrary, our NAS focuses creating an efficient training model zoo for fast fine-tuning. Moreover, the existing search space design cannot meet the purpose of our search.

**Generating Hyperparameters for Fine-tuning.** HPO methods such as Bayesian optimization (BO) (Strubell et al., 2019a; Mendoza et al., 2016), BOHB (Falkner et al., 2018) achieves very promising result but require a lot of computational resources which is contradictory to our original

Figure 1: Overview of our NASOA. Our faster task-oriented online fine-tuning system has two parts: a) Offline NAS to generate an efficient training model zoo with good accuracy and training speed; b) An online fine-tuning regime generator to perform a task-specific fine-tuning with a suitable model under user's time constraint.

objective of efficient fine-tuning. On the other hand, limited works discuss the model selection and HPO for fine-tuning. Kornblith et al. (2019) finds that ImageNet accuracy and fine-tuning accuracy of different models are highly correlated. Li et al. (2020); Achille et al. (2019) suggest that the optimal hyperparameters and model for fine-tuning should be both dataset dependent and domain similarity dependent (Cui et al., 2018). HyperStar (Mittal et al., 2020) is a concurrent HPO work demonstrating that a performance predictor can effectively generate good hyper-parameters for a single model. However, those works don't give an explicit solution about how to perform fine-tuning in a more practical online scenario. In this work, we take the advantage of online learning (Hoi et al., 2018; Sahoo et al., 2017) to build a schedule generator, which allows us to memorize the past training history and provide up-and-coming training regimes for new coming tasks on the fly. Besides, we introduce the NAS model zoo to further push up the speed and performance.

## 3 THE PROPOSED APPROACH

The goal of this paper is to develop an online fine-tuning pipeline to facilitate a fast continuous cross-task model adaption. By the preliminary experiments in Section 4.1, we confirm that the model architectures and hyperparameters such as the learning rate and frozen stages will greatly influence the accuracy and speed of the fine-tuning program. Thus, our NASOA includes two parts as shown in the Figure 1: 1) Searching a group of neural architectures with good accuracy and fast training speed to create a pretrained model zoo; 2) Designing an online task-oriented algorithm to generate an efficient fine-tuning regime with the most suitable model under user's time constraint.

### 3.1 CREATING AN EFFICIENT TRAINING MODEL ZOO (ET-NAS) BY NAS

The commonly used hand-craft backbones for fine-tuning including MobileNet (Sandler et al., 2018), ResNet (He et al., 2016), and ResNeXt (Xie et al., 2017). Recently, some state-of-the-art backbone series such as RegNet (Radosavovic et al., 2020), and EfficientNet (Tan et al., 2019b) are developed by automated algorithms for higher accuracy and faster inference speed. However, the objective of our NAS is to find a group of models with good model generalization ability and training speed. Suggested by Kornblith et al. (2019), the model fine-tuning accuracy (model generalization ability) has a strong correlation between ImageNet accuracy ($r = 0.96$). Meanwhile, the training speed can be measured by the step time of each training iteration. Thus, our NAS can be formulated by a multi-objective optimization problem (MOOP) on the search space $S$ given by:

$$\max_{\mathcal{A} \in S} \left( \text{acc}(\mathcal{A}), -T_s(\mathcal{A}) \right) \text{ subject to } T_s(\mathcal{A}) \leq T_m \tag{1}$$

where $\mathcal{A}$ is the architecture, $\text{acc}(.)$ is the Top-1 accuracy on ImageNet, $T_s(.)$ is the average step time of one iteration, and $T_m$ is the maximum step time allowed. The step time is defined to be the total time of one iteration, including forward/backward propagation, and parameter update.

**Search Space Design** is extremely important (Radosavovic et al., 2020). As shown in Figure 2, to enable an efficient model, we propose a novel flexible joint block-level and macro-level search space to enable simple to complex block design and fine adjustment of the computation allocation on each stage. Unlike existing topological cell-level search space such as DARTS (Liu et al., 2018b), AmoebaNet(Real et al., 2019a), and NASBench101(Dong & Yang, 2019), ours is more compact and avoids redundant skip-connections which have great memory access cost (MAC). Our block-level search space is more flexible to adjust the width, depth (for each stage), when to down-sample/raise the channels. In contrast, EfficientNet only scales up/down the total width and depth by a fixed allocation ratio, and RegNet cannot change the number/type of operations in each block.

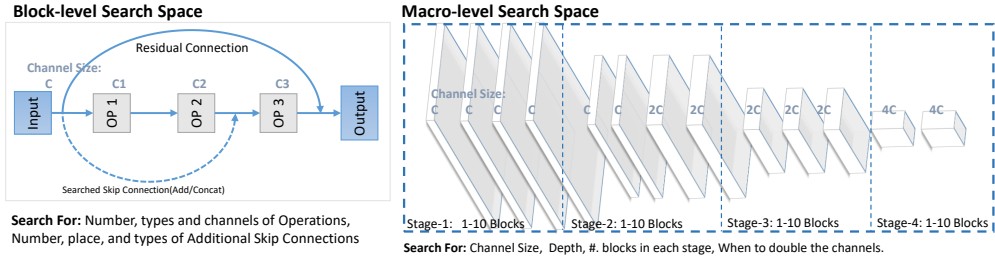

Figure 2: Our joint block/macro-level search space to find efficient training networks. Our block-level search space covers many popular designs such as ResNet, ResNext, MobileNet Block. Our macro-level search space allows small adjustment of the network in each stage thus the resulting models are more flexible and efficient.

**Block-level Search Space.** We consider a search space based on 1-3 successive nodes of 5 different operations. Three skip connections with one fixed residual connection are searched. Element-wise add or channel-wise concat is chosen to combine the features for the skip-connections. For each selected operation, we also search for the ratio of changing channel size: $\times 0.25, \times 0.5, \times 1, \times 2, \times 4$. Note that it can cover many popular block designs such as Bottleneck (He et al., 2016), ResNeXt (Xie et al., 2017) and MB block (Sandler et al., 2018). It consists of $5.4 \times 10^6$ unique blocks.

**Macro-level Search Space.** Allocation computation over different stages is crucial for a backbone (Liang et al., 2020). Early-stage feature maps in one backbone are larger which captures texture details, while late-stage feature maps are smaller which are more discriminative (Li et al., 2018). Therefore, for macro-level search space, we design a flexible search space to find the optimal channel size (width), depth (total number of blocks), when to down-sample, and when to raise the channels. Our macro-level structure consists of 4 flexible stages. The spatial size of the stages is gradually down-sampled with factor 2. In each stage, we stack a number of block architectures. The positions of the doubling channel block are also flexible. This search space consists of $1.5 \times 10^7$ unique architectures. Details of the search space and its encodings can be found in Appendix B.1.

**Multi-objective Searching Algorithm**. For MOOP in Eq 1, we define architecture $\mathcal{A}_1$ *dominates* $\mathcal{A}_2$ if (i) $\mathcal{A}_1$ is no worse than $\mathcal{A}_2$ in all objectives; (ii) $\mathcal{A}_1$ is strictly better than $\mathcal{A}_2$ in at least one objective. $\mathcal{A}^*$ is *Pareto optimal* if there is no other $\mathcal{A}$ that dominate $\mathcal{A}^*$. The set of all *Pareto optimal* architectures constitutes the *Pareto front*. To solve this MOOP problem, we modify a well-known method named Elitist Non-Dominated sorting genetic algorithm (NSGA-II) (Deb et al., 2000) to optimize the *Pareto front* $\mathcal{P}_f$. The main idea of NSGA-II is to rank the sampled architectures by non-dominated sorting and preserve a group of elite architectures. Then a group of new architectures is sampled and trained by mutation of the current elite architectures on the $\mathcal{P}_f$. The algorithm can be paralleled on multiple computation nodes and lift the $\mathcal{P}_f$ simultaneously. We modify the NSGA-II algorithm to become a NAS algorithm: a) To enable parallel searching on N computational nodes, we modify the non-dominated-sort method to generate exactly N mutated models for each generation, instead of a variable size as the original NSGA-II does. b) We define a group of mutation operations for our block/macro search space for NSGA-II to change the network structure dynamically. c) We add a parent computation node to measure the selected architecture's training speed and generate the Pareto optimal models. Details of our NSGA-II can be found in Appendix B.2.2.

**Efficient Training Model Zoo** $Z_{oo}$ **(ET-NAS).** By the proposed NAS method, we then create an efficient-training model zoo $Z_{oo}$ named ET-NAS which consists of $K$ *Pareto optimal* models $\mathcal{A}_i^*$ on $\mathcal{P}_f$. Then $\mathcal{A}_i^*$ are pretrained by ImageNet. Details of our NAS and $\mathcal{A}_i^*$ can be found in Appendix B.

### 3.2 ONLINE TASK-ORIENTED FINE-TUNING SCHEDULE GENERATION

With the help of efficient training $Z_{oo}$, the marginal computational cost of each user is minimized while they can enjoy the benefit of NAS. We then need to decide a suitable fine-tuning schedule upon the user's upcoming tasks. Given user's dataset $D$ and fine-tuning time constraint $T_l$, an online regime generator $G(.,.)$ is desired:

$$[\text{Regime}_{\text{FT}}, \mathcal{A}_i^*] = G(D, T_l), \text{ such that } \text{Acc}(\mathcal{A}_i^{FineTune}, D_{val}) \text{ is maximized,} \qquad (2)$$

where the $\text{Regime}_{\text{FT}}$ includes all the hyperparameters required, i.e., $lr$ schedule, total training steps, and frozen stages. $G(.,.)$ also needs to pick up the most suitable pretrained model $\mathcal{A}_i^*$ from $Z_{oo}$. Note that existing search-based HPO methods require huge computational resources and cannot fit

in our online one-shot training scenario. Instead, we first propose an online learning predictor $Acc_P$ to model the accuracy on the validation set $\mathrm{Acc}(\mathcal{A}_i^{FT}, D_{val})$ by the meta-data information. Then we can use the predictor to construct $G(.,.)$ to generate an optimal hyperparameter setting and model.

### 3.2.1 ONLINE LEARNING FOR MODELING $\mathrm{Acc}(\mathcal{A}_i^{FT}, D_{val})$

Recently, Li et al. (2020) suggest that the optimal hyperparameters for fine-tuning are highly related to some data statistics such as domain similarity to the ImageNet. Thus, we hypothesis that we can model the final accuracy by a group of predictors, e.g., model information, meta-data description, data statistics $\mathrm{stat}(D)$, domain similarity, and hyperparameters. We list the variables we considered to predict the accuracy result as follows:

| Model $\mathcal{A}_i^*$ name (one-hot dummy variable) | | ImageNet Acc. of the $\mathcal{A}_i^*$ | |
| --- | --- | --- | --- |
| Domain Similiarity to ImageNet (EMD) (Cui et al., 2018) | | #.Classes | Number of Iteration |
| Average #. images per class | Std #. images per class | Learning Rate | Frozen Stages |

Those variables can be easily calculated ahead of the fine-tuning. One can prepare offline training data by fine-tuning different kinds of dataset and collect the accuracy correspondingly and apply a Multi-layer Perceptron Regression (MLP) offline on it. However, online learning should be a more realistic setting for our problem. In cloud computing service or a GPU cluster, a sequence of fine-tuning requests with different data will arrive from time to time. The predictive model can be further improved by increasing the diversity of the data and the requests over time.

Using a fixed depth of MLP model in the online setting may be problematic. Shallow networks maybe more preferred for small number of instances, while deeper model can achieve better performance when the sample size becomes larger. Inspired by Sahoo et al. (2017), we use an adaptive MLP regression to automatically adapt its model capacity from simple to complex over time. Given the input variables, the prediction of the accuracy is given by:

$$Acc_P(\mathcal{A}_i^*, \mathrm{Regime_{FT}}, stat(D)) = \sum_{l=1}^{L} \alpha_l \mathrm{f}_l, \text{ where } l = 0, ..., L \tag{3}$$

$$\mathrm{f}_l = h_l W_l, \ h_l = RELU(\Phi_l h_{l-1}), \ h_0 = [\mathcal{A}_i^*, \mathrm{Regime_{FT}}, stat(D)].$$

The predicted accuracy is a weighted sum of the output $\mathrm{f}_l$ of each intermediate fully-connected layer $h_l$. The $W_l$ and $\Phi_l$ are the learnable weights of each fully-connected layer. The $\alpha_l$ is a weight vector assigning the importance to each layer and $\|\alpha\| = 1$. Thus the predictor $Acc_P$ can automatically adapt its model capacity from simple to complex along with incoming tasks. The learnable weight $\alpha_l$ controls the importance of each intermediate layer and the final predicted accuracy is a weighted sum of $\mathrm{f}_l$ of them. The network can be updated by a Hedge Backpropagation (Freund & Schapire, 1999) in which $\alpha_l$ is updated based on the loss suffered by this layer $l$ as follows:

$$\alpha_l' \leftarrow \alpha_l \beta^{\mathcal{L}(\mathrm{f}_l, Acc_{gt})}, \ W_l' \leftarrow W_l - \eta \alpha_l \nabla_{W_l} \mathcal{L}(\mathrm{f}_l, Acc_{gt})$$

$$\Phi_l' \leftarrow \Phi_l' - \eta \sum_{j=l}^{L} \alpha_j \nabla_{W_l} \mathcal{L}(\mathrm{f}_j, Acc_{gt}), \ \alpha_l'' \leftarrow \frac{\alpha_l'}{\sum \alpha_l'}$$

where $\beta \in (0, 1)$ is the discount rate, the weight $\alpha_l'$ are re-normalized such that $\|\alpha\| = 1$, and $\eta$ is the learning rate. Thus, during the online update, the model can choose an appropriate depth by $\alpha_l$ based on the performance of each output at that depth. By utilizing the online cumulative results, our generator gains experience that helps future prediction.

**Generating Task-oriented Fine-tuning Schedule.** Our schedule generator $G$ then can make use of the performance predictor to find the best training regime:

$$G(D, T_l) = \arg\max_{\mathcal{A} \in Z_{oo}, Regime_{FT} \in S_{FT}} Acc_P(\mathcal{A}, Regime_{FT}, \mathrm{stat}(D)).$$

Once the time constraint $T_l$ is provided, the max number of iterations for different $\mathcal{A}_i^*$ can be calculated by an offline step-time lookup table for $Z_{oo}$. The corresponding meta-data variables can be then calculated for the incoming task. The optimal selection of model and hyperparameters is obtained by ranking the predicted accuracy of all possible grid combinations. The Detailed algorithm can be found in the Appendix B.6.

Table 1: Datasets and their statistics used in this paper. Datasets in bold are used to construct the online learning training set. The rest are used to test our NASOA. It is commonly believed that Aircrafts, Flowers102 and Blood-cell deviate from the ImageNet domain.

| DataSets | #.Class | Task | #.Train | #.Test | DataSets | #.Class | Task | #.Train | #.Test |
|---|---|---|---|---|---|---|---|---|---|
| **Flowers102**(Nilsback & Zisserman, 2008) | 102 | Fine-Grained | 6K | 2K | Stanford-Car(Krause et al., 2013) | 196 | Fine-Grained | 8K | 8K |
| **CUB-Birds**(Welinder et al., 2010) | 200 | Fine-Grained | 10K | 2K | MIT67(Quattoni & Torralba, 2009) | 67 | Scene cls. | 5K | 1K |
| **Caltech101**(Fei-Fei et al., 2006) | 101 | General | 8K | 1K | Food101(Bossard et al., 2014) | 101 | Fine-Grained | 75K | 25K |
| **Caltech256**(Griffin et al., 2007) | 257 | General | 25K | 6K | FGVC Aircrafts(Maji et al., 2013) | 100 | Fine-Grained | 7K | 3K |
| **Stanford-Dog**(Khosla et al., 2011) | 120 | Fine-Grained | 12K | 8K | Blood-cell(Singh et al., 2020) | 4 | Medical Img. | 10K | 2K |

## 4 EXPERIMENTAL RESULTS

### 4.1 PRELIMINARY EXPERIMENTS

We conduct a complete preliminary experiment to justify our motivation and model settings. Details can be found in the Appendix A. According to our experiments, we find that for an efficient fine-tuning, the model matters most. **The suitable model should be selected according to the task and time constraints**. Thus constructing a model zoo with various sizes of training-efficient models and picking up suitable models should be a good solution for faster fine-tuning. We also verify some existing conclusions: Fine-tuning performs better than training from scratch (Kornblith et al., 2019) so that our topic is very important for efficient GPU training; Learning rate and frozen stage are crucial for fine-tuning (Guo et al., 2019), which needs careful adjustment.

### 4.2 OFFLINE NAS AND MODEL ZOO RESULTS

During the NAS, we directly search on the ImageNet dataset(Russakovsky et al., 2015). We first search for a group of efficient block structure, then use those block candidates to conduct the macro-level search. We use a short training setting to evaluate each architecture. It takes about 1 hour on average for evaluating one architecture for the block-level search and 6 hours for the macro-level search. Paralleled on GPUs, it takes about one week on a 64-GPU cluster to conduct the whole search (5K+1K arch). Implementation details and intermediate results can be found in the Appendix B.

**Faster Fine-tuning Model Zoo (ET-NAS).** After identifying the $\mathcal{A}_i^*$ from our search, we fully train those models on ImageNet following common practice. Note that all the models including ET-NAS-L can be easily pretrained on a regular 8-card GPU node since our model is training-efficient. **We are confident to release our models for the public to reproduce our results from scratch**[2] **and let the public to save their energy/CO2/cost**. Due to the length of the paper, we put the detailed encoding and architectures of the final searched models in the Appendix B.4. Surprisingly, we found that smaller models should use simpler structures of blocks while bigger models prefer complex blocks. Comparing our searched backbone to the conventional ResNet/ResNeXt, we find that early stages in our models are very short which is more efficient since feature maps in an early stage are very large and the computational cost is comparably large. This also verified our findings in Appendix B.4.1.

Table 2: Comparsion of our ET-NAS models and SOTA ImageNet models. Inference time and training step time are measured in ms on single Nvidia V100, with $bs = 64$.

| Model Name | Top-1 Acc. | Inf Time (ms) | Training Step Time (ms) |
|---|---|---|---|
| RegNetY-200MF | 70.40 | 14.25 | 62.30 |
| **ET-NAS-C** | **71.29** | **8.94** | **26.28** |
| RegNetY-400MF | 74.10 | 20.57 | 90.61 |
| **ET-NAS-D** | **74.46** | **14.54** | **36.30** |
| RegNetY-600MF | 75.50 | 22.15 | 90.11 |
| MobileNet-V3-Large | 75.20 | **16.88** | 71.65 |
| OFANet | 76.10 | 17.81 | 73.10 |
| Amoebanet | 75.70 | 28.39 | 141.45 |
| **ET-NAS-E** | **76.87** | 25.34 | **61.95** |
| EfficientNet-B0 | 77.70 | 24.30 | 120.29 |
| RegNetY-800MF | 78.00 | 45.59 | 170.96 |
| **ET-NAS-F** | **78.80** | **33.83** | **93.04** |
| EfficientNet-B2 | 80.40 | 58.78 | 277.60 |
| RegNetY-16GF | 80.40 | 192.78 | 677.68 |
| **ET-NAS-G** | **80.41** | **53.08** | **133.97** |
| **ET-NAS-H** | **80.92** | 76.80 | 193.40 |
| EfficientNet-B3 | 81.50 | 97.33 | 455.86 |
| **ET-NAS-I** | 81.38 | **94.60** | **265.13** |
| **ET-NAS-J** | **82.08** | 131.92 | 370.28 |
| **ET-NAS-L** | **82.65** | 191.89 | 542.52 |

**Comparison with the state-of-the-art ImageNet models.** We compare the training/inference efficiency of our searched ET-NAS with the SOTA ImageNet models such as MobileNetV3 (Howard et al., 2019), RegNet series (Radosavovic et al., 2020), and EfficientNet series (Tan & Le, 2019)

---

[2]The efficient training model zoo (ET-NAS) has been released at: `https://github.com/NAS-OA/NASOA`

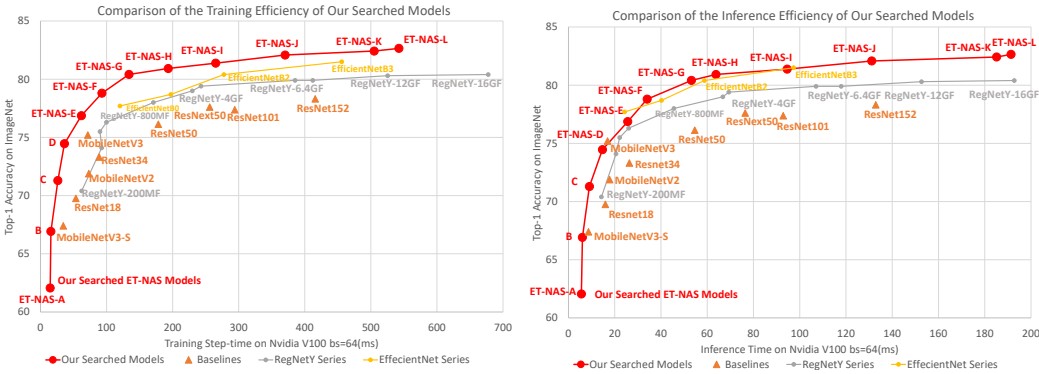

Figure 3: Comparison of the training and inference efficiency of our searched models (ET-NAS) with SOTA models on ImageNet. Our searched models are considerably faster, e.g., ET-NAS-G is 6x training faster than RegNetY-16GF, and ET-NAS-I is 1.5x training faster than EfficientNetB3. Although our models are optimized for fast training, the inference speed is comparable to EfficientNet and better than RegNet series.

as shown in Table 2 and Figure 3. Overall, our searched models outperform other SOTA ImageNet models in terms of training accuracy and training speed from Figure 3 (left). Specifically, ET-NAS-G is about 6x training faster than RegNetY-16GF, and ET-NAS-I is about 1.5x training faster than EfficientNetB3. Our models are also better than mobile setting models such as MobileNetV2/V3(Howard et al., 2019) and RegNetY-200MF. Although our model is optimized for fast training, we also compare the inference speed in Figure 3(right). Our models still have a very strong performance in terms of inference speed, outperforming RegNet series and achieving comparable performance with EfficientNet. In Figure 3 (mid), our method is more training efficient than other NAS results, e.g.some evolution-based NAS methods such as AmoebaNet, OFANet.

**What makes our network training efficient?**
To answer this, we define an *efficiency score* and conduct a statistical analysis of different factors for efficient-training (Details can be found in Appendix B.4.1). We have the following conclusions: a) By observing optimal $\mathcal{A}_i^*$, smaller models should use simpler blocks while bigger models prefer complex blocks. Using the same block structure for all sizes of models (Tan & Le, 2019; Radosavovic et al., 2020) may not be optimal. b) Adding redundant skip-connections which have great memory access cost will decrease the training efficiency of the model thus existing topological cell-level search space such as DARTS (Liu et al., 2018b), AmoebaNet (Real et al., 2019a), and NASBench101 (Dong & Yang, 2019) is not efficient. c) The computation allocation on different stages is crucially important. Simply increasing depth/width to expand the model as Tan & Le (2019) may not be optimal will downgrade the performance. To conclude our novel joint search space contributes most to the training efficiency.

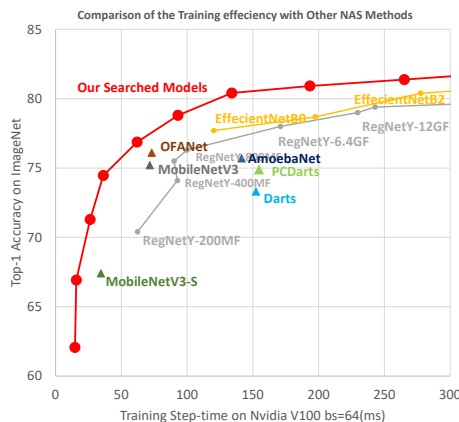

Figure 4: Comparison of the training efficiency of our searched models (ET-NAS) with 8 other NAS results on ImageNet. It can be found that our method is more training efficient than some recent evolution-based NAS methods such as AmoebaNet, OFANet because of our effective search space.

### 4.3 RESULTS FOR ONLINE ADAPTIVE PREDICTOR $Acc_P$

**Experimental Settings**. We evaluate our online algorithm based on ten widely used image classification datasets, that cover various fine-tuning tasks as shown in Table 1. Five of them (in bold) are chosen to be the online learning training set (meta-training dataset). 30K samples are collected by continually sampling a subset of each dataset and fine-tuning with a randomized hyperparameters on it. Each subset varies from #. classes and #. images. The variables in Section 3.2.1 are calculated

accordingly. The fine-tuning accuracy is then evaluated on the test set. Then 30K sample is split into 24K meta-training samples and 6K meta-validation samples.

Then an adaptive MLP regression in Eq 3 are used to fit the data and predict the $Acc(\mathcal{A}_i^{FT}, D_{val})$. We use $L = 10$ with 64 units in each hidden layer. We use a learning rate of 0.01 and $\beta = 0.99$. As baselines, we also compare the results of using fixed MLP with plain backpropagation with different layers($L = 3, 6, 10, 14$). MAE (mean absolute error) and MSE (mean square error) are performance metrics to measure the cumulative error with different segments of the tasks stream.

Table 3: Online error rate of our method and fixed MLP. Our adaptive MLP with hedge backpropagation is better in the online setting of predicting the fine-tuning accuracy.

| Models | All Cumulative Err. | | Segment 20-40% | | Segment 80-100% | |
|---|---|---|---|---|---|---|
| | MAE | MSE | MAE | MSE | MAE | MSE |
| Fixed MLP (L=3) | 10.07% | 1.94% | 8.99% | 1.56% | 7.99% | 1.21% |
| Fixed MLP (L=6) | 9.12% | 1.71% | 9.03% | 1.62% | 7.16% | 1.04% |
| Fixed MLP (L=10) | 8.45% | 1.59% | 8.46% | 1.53% | 6.68% | 0.96% |
| Fixed MLP (L=14) | 11.24% | 2.91% | 8.34% | 1.54% | 4.62% | 0.46% |
| Adaptive MLP w Hedge-BP | **7.51%** | **1.36%** | **7.55%** | **1.11%** | **3.73%** | **0.28%** |

**Comparison of online learning method.** The cumulative error obtained by all the baselines and the proposed method to predict the fine-tuning accuracy is shown in Table 4. It can be seen that our adaptive MLP with hedge backpropagation is better than fixed MLP in terms of the cumulative error of the predicted accuracy. Our method enjoys the benefit from the adaptive depth which allows faster convergence in the initial stage and strong predictive power in the later stage.

## 4.4 FINAL NASOA RESULTS

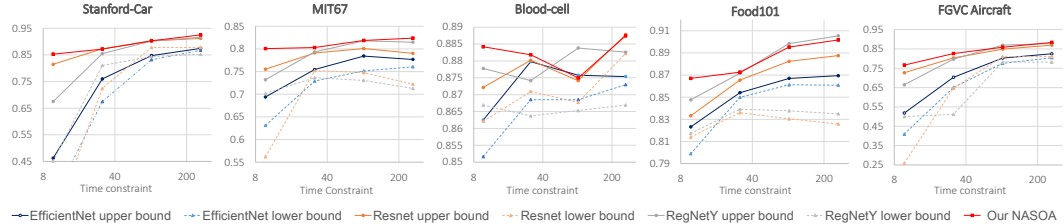

Figure 5: Comparison of the final fine-tuning results under four time constraints for the testing dataset. Red square lines are the results of our NASOA in one-shot. The dots on the other solid line are the best performance of all the models in that series can perform. The model and training regime generated by our NASOA can outperform the upper bound of other methods in most cases. Our methods can improve around 2.1%~7.4% accuracy than the upper bound of RegNet/EfficientNet series on average.

To evaluate the performance of our whole NASOA, we select four time constraints on the testing datasets and use $Acc_P(.)$ to test the fine-tuning accuracy. The testing datasets are MIT67, Food101, Aircrafts, Blood-cell, and Stanford-Car. The shortest/longest time constraint are the time of fine-tuning 10/50 epochs for ResNet18/ResNet101. The rest are equally divided into the log-space. For our NASOA, we generate the fine-tuning schedules by maximizing the predicted accuracy in Eq 2. We also conduct fine-tuning on various candidates of baselines such as ResNet (R18 to R152), RegNet (200MF to 16GF), and EfficientNet (B0 to B3) with the default hyperparameter setting in Li et al. (2020).

**Comparison of the final fine-tuning results with the SOTA networks.** We plot the time versus accuracy comparison in Figure 5. As can be seen, the model and

Table 4: Comparison of the final NASOA results with other HPO methods. "HPO only" means only optimizing the hyperparameters with RegNetY-16GF. Other HPO methods optimize both selecting hyperparameters and model from RegNet series models. "OA only" is our online schedule generator with RegNet series models. "Our $Z_{oo}$" means using our ET models zoo to find suitable model. "Fixed MLP Predictor" is the offline baseline with fixed MLP predictor. "Our NASOA" is the our whole pipeline with both training efficient model zoo and online adaptive scheduler. Without additional search cost (x40), NASOA can reach similar performance of BOHB.

| Methods | Search Cost | Aircrafts | MIT67 | Sf-Car | Sf-Dog |
|---|---|---|---|---|---|
| Random Search (HPO only) | x40 | 63.07% | 75.60% | 67.47% | 86.25% |
| BOHB (HPO only) | x40 | 72.70% | 77.61% | 70.94% | 87.41% |
| Random Search | x40 | 81.07% | 79.93% | 88.99% | 89.06% |
| BOHB | x40 | **82.34%** | **79.85%** | **89.01%** | **89.49%** |
| Our $Z_{oo}$ with Random Search | x40 | 83.71% | 80.97% | 87.84% | 92.75% |
| Our $Z_{oo}$ with BOHB | x40 | 84.67% | 82.34% | 89.03% | 93.74% |
| Our OA only | x1 | 81.22% | 79.33% | 84.56% | 89.70% |
| Our $Z_{oo}$ with Fixed MLP Predictor (Offline) | x1 | 81.31% | 75.97% | 88.81% | 88.58% |
| **Our final NASOA** | **x1** | **82.54%** | **80.30%** | **88.20%** | **92.30%** |

training regime generated by our NASOA can outperform the upper bound of other methods in most cases. On average, our methods can improve around 2.1%/7.4% accuracy than the best model of RegNet/EfficientNet series under various time constraints and tasks. It is noteworthy that our NASOA performs better especially in the case of short time constraint, which demonstrates that our schedule generator is capable provide both efficient and effective regimes for fast fine-tuning.

**Comparison of the final fine-tuning results with the HPO methods.** In Table 4, we compare our method with the HPO methods which optimizing the hyperparameters and picking up models in ResNet, RegNetY, and EfficientNet series. "HPO only" means the method only optimizes the hyperparameters with a fixed model RegNetY-16GF. "OA only" is our online schedule generator with RegNet series models. "Our $Z_{oo}$" means using our ET models zoo to find suitable model. "Fixed MLP Predictor" is the offline baseline with fixed MLP predictor (L=10) with our model zoo. "Our NASOA" is the our whole pipeline with both training effi-

Table 5: . This ablative study calculates the average fine-tuning accuracy over 5 tasks.

| Methods | Fixed Model | Existing Models | NAS Models | Adaptive Scheduler | Comp Cost | Avg. Fine-tuning Accuracy |
|---|---|---|---|---|---|---|
| BOHB[10] | ✓ | | | | x40 | 77.17% |
| + Our Zoo | | | ✓ | | x40$^{-0\times}$ | 87.45% $^{+10.28\%}$ |
| Our OA | | ✓ | | ✓ | x1$^{-40\times}$ | 83.70% $^{+6.54\%}$ |
| NASOA | | | ✓ | ✓ | x1$^{-40\times}$ | 85.84% $^{+8.67\%}$ |

cient model zoo and online adaptive scheduler. Comparing to the offline baseline with our NASOA, our online adaption module can boost the average performance by 2.17%. It can also be found that our method can save up to 40x computational cost compared to HPO methods while reaching similar performance. With more computational budget, our model zoo with BOHB search can reach even higher accuracy (+avg. 10.28%).

**Ablative interpretation of performance superiority.** Table 5 calculates the average fine-tuning accuracy over tasks. Our NAS model zoo can greatly increase the fine-tuning average accuracy from 77.17% to 87.45%, which is the main contribution of the performance superiority. Using our online adaptive scheduler instead of BOHB can significantly reduce the computational cost (-40x).

## 5    CONCLUSION

We propose the first efficient task-oriented fine-tuning framework aiming at saving the resources for GPU clusters and cloud computing. The joint NAS and online adaption strategy achieve much better fine-tuning results in terms of both accuracy and speed. The searched architectures are more training-efficient than very strong baselines such as RegNet and EfficientNet. Our experiments on multiple datasets show our NASOA achieves 40x speed-up comparing to BOHB. The proposed NASOA can be well adapted to more tasks such as detection and segmentation in the future.

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

## A  PRELIMINARY EXPERIMENTS

### A.1  EXPERIMENTS SETTINGS

The preliminary experiments aim at figuring out what kinds of factors impact the speed and accuracy of fine-tuning. We fine-tune several ImageNet pretrained backbones on various datasets as shown in Table 6 (right) and exam different settings of hyperparameters by a grid search such as: learning rate (0.0001, 0.001, 0.01, 0.1), frozen stages (-1,0,1,2,3), and frozen BN (-1,0,1,2,3). Frozen stages/frozen BN= $k$ means 1 to $k$th stage's parameters/BN statistics are not updated during training. The training settings most follow Li et al. (2020) and we report the Top-1 validation accuracy and training time. Its detailed experiment settings are hyperparameters are listed as follows:

**Comparing fine-tuning and training from scratch**. We use ResNet series (R-18 to R-50) to evaluate the effect of fine-tuning and training from scratch. Following Li et al. (2020), we train networks on Flowers102, CUB-Birds, MIT67 and Caltech101 datasets for 600 epochs for training from scratch and 350 epochs for fine-tuning to ensure all models converge on all datasets. We use SGD optimizer with an initial learning rate 0.01, weight decay 1e-4, momentum 0.9. The learning rate is decreased by factor 10 at 400 and 550 epoch for training from scratch and 150, 250 epoch for fine-tuning.

**Optimal learning rate and frozen stage**. We perform a simple grid search on Flowers102, Stanford-Car, CUB-Birds, MIT67, Stanford-Dog, and Caltech101 datasets with ResNet50 to find optimal learning rate and frozen stage on different datasets with the default fine-tune setting in Li et al. (2020). The hyperparameters ranges are: learning rate (0.1, 0.01, 0.001, 0.0001), frozen stage (-1, 0, 1, 2, 3).

**Comparing different frozen stages and networks along time**. We fix different stages of ResNet50 to analyze the influence of different frozen stages to the accuracy and along the training time on Flowers102, Stanford-Car, CUB-Birds, MIT67, Stanford-Dog, and Caltech101 datasets. We pick

Table 6: Comparison of Top-1 accuracy and training time (min) on different datasets. Comparing to training from scratch, fine-tuning shows superior results in terms of both accuracy and training time.

| Dataset | Method | ResNet18 | | ResNet34 | | ResNet50 | |
|---|---|---|---|---|---|---|---|
| | | Acc. | Time (min) | Acc. | Time (min) | Acc. | Time (min) |
| Flowers102 | From Scratch | 94.4% | 11 | 93.8% | 19 | 90.7% | 38 |
| | Fine-tuning | **98.3%** | **7** | **98.5%** | **11** | **98.8%** | **22** |
| Stanford-Car | From Scratch | 80.6% | 14 | 81.9% | 24 | 81.5% | 47 |
| | Fine-tuning | **87.6%** | **8** | **89.6%** | **14** | **91.1%** | **28** |
| CUB-Birds | From Scratch | 51.6% | 11 | 53.6% | 18 | 44.6% | 35 |
| | Fine-tuning | **69.2%** | **6** | **71.9%** | **10** | **74.9%** | **21** |
| MIT67 | From Scratch | 67.8% | 22 | 69.2% | 37 | 66.0% | 73 |
| | Fine-tuning | **76.4%** | **13** | **76.9%** | **21** | **78.1%** | **43** |
| Stanford-Dog | From Scratch | 60.6% | 21 | 62.6% | 35 | 55.2% | 70 |
| | Fine-tuning | **69.7%** | **12** | **73.3%** | **20** | **75.0%** | **41** |
| Caltech101 | From Scratch | 82.5% | 12 | 78.1% | 20 | 75.7% | 40 |
| | Fine-tuning | **90.8%** | **7** | **91.8%** | **12** | **91.8%** | **23** |

Table 7: Fine-tuning on R50, the optimal learning rate and optimal frozen stage found by grid search are different and should be optimized individually.

| DataSet | #.Class | #.Images | Optimal LR | Optimal Frozen Stage | Best Acc. |
|---|---|---|---|---|---|
| Flowers102 | 102 | 8K | 0.01 | 0 | 99.3% |
| Stanford-Car | 196 | 16K | 0.1 | 1 | 91.8% |
| CUB-Birds | 200 | 12K | 0.01 | 2 | 81.3% |
| MIT67 | 67 | 16K | 0.01 | -1 | 80.8% |
| Stanford-Dog | 120 | 21K | 0.01 | 3 | 83.7% |
| Caltech101 | 101 | 8K | 0.001 | -1 | 96.4% |
| Caltech256 | 257 | 31K | 0.01 | 3 | 85.6% |

the training curves on CUB-Birds and Caltech101 to in the main text of this paper. We also compare the fine-tune results along time with various networks on these datasets as shown in Figure 7. On Caltech101, ResNet50 dominates the training curve at the very beginning. However, on other datasets, ResNet18 and ResNet34 can perform better then ResNet50 when the training time is short.

## A.2 FINDINGS OF THE PRELIMINARY EXPERIMENTS

With those preliminary experiments, we summarize our findings as follows. Some of the findings are also verified by some existing works.

- **Fine-tuning performs always better than training from scratch.** As shown in Table 6, fine-tuning shows superior results than training from scratch in terms of both accuracy and training time for all the datasets. This finding is also verified by Kornblith et al. (2019). Thus, fine-tuning is the most common way to train a new dataset and our framework can be generalized to applications.

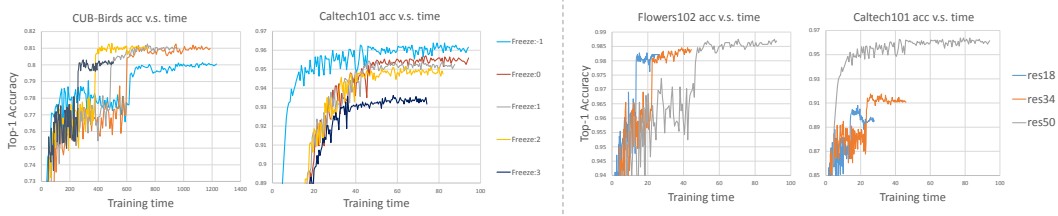

Figure 6: (Left) Fine-tuning ResNet101 with different weight-frozen stages. "Freeze: k" means 0 to k stage's parameters are not updated during training. The number of frozen stage will effect both training time and accuracy. Its optimal frozen setting varies with datasets. (Right) Comparison of accuracy/time different fine-tuning models. Different models should be selected upon the request of different datasets and training constraints.

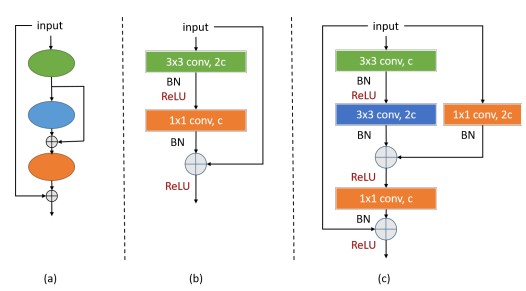

Figure 7: Fine-tune results along time with various networks on these datasets. It can be seen that if the time constraints is short, we should choose a smaller network.

- **We should the optimize learning rate and frozen stage for each dataset.** From Table 7, it seems that the optimal learning rate and optimal frozen stage found by grid search are different for various datasets. Figure 6also shows that the number of the frozen stages will affect both training time and final accuracy. Guo et al. (2019) also showed that frozen different stages are crucial for fine-tuning task. Those two hyperparameters should be optimized for different datasets.

- **Model matters most. Suitable model should be selected according to the task and time constraints.** Figure 6 (right) suggests that always choosing the biggest model to fine-tune may not be an optimal choice, smaller model can be better than the bigger model if the training time is limited. On the other hand, it is also important to consider the training efficiency of the model since a better model can be converged faster by a limited GPU budget. For example, Figure 7 shows that if the time constraint is short, we should choose a smaller network i.e. ResNet18 here. Thus, it is urgent to construct a training-efficient model zoo.

- **BN running statistics should not be frozen during fine-tuning.** We found that frozen BN has a very limited effect on the training time (less than $\pm 5\%$), while not freezing BN will lead to better results in almost all the datasets. Thus, BN is not frozen all experiments for our NASOA.

## B   DETAILS OF THE OFFLINE NAS

### B.1   SEARCH SPACE ENCODINGS

The search space of our architectures is composed of block and macro levels, where the former decides what a block is composed of, such as operators, number of channels, and skip connections, while the latter is concerned about how to combine the block into a whole network, e.g., when to do down-sampling, and where to change the number of channels.

### B.1.1   BLOCK-LEVEL ARCHITECTURE

**Block-level design.** A block consists of at most three operators, each of which is divided into 5 species and has 5 different number of output channels. Each kind of operator is denoted by an op number, and the output channel of the operator is decided by the ratio between it and that of the current block. Details are shown in Table B.1.1. By default, there is a skip connection between the input and output of the block, which sums their values up. In addition to that, at most 3 other skip connections are contained in a block, which either adds or concatenates the values between them. Each operation is followed by a batch normalization layer, and after all the skip connections are calculated, a ReLU layer is triggered.

Figure 8: Block structure and two block samples. (a) shows a three-node graph. (b) is an example with encoding "031-", and (c) is "02031-a02".

**Block-level encoding.** The encoding of each block-level architecture is composed of two parts separated by '-', which considers the operators and skip connections respectively.

For the first part (operators part), each operator is represented by two numbers: op number and ratio number (shown in Table B.1.1). As the output channel of the last operation always equals to that of the current block, the ratio number of this operator is removed. Therefore, the encoding of a block with $n$ operators always has length $2n - 1$ for the first part of block-level encoding.

For the second part (skip connections part), every skip connection consists of one letter for addition ('a') / concatenation ('c'), and two numbers for place indices. $n$ operators separate the block to $n+1$ parts, which are indexed with $0, 1, 2, \ldots, n$. Thus 'a01' means summing up the value before and after the first operator. Since the skip connection between the beginning and end of the block always exists, it is not shown in the encoding. Thus this part has length $3k - 3$ (possibly 0) when there is $k$ skip connections. Some of the encoding examples are shown in Figure 8.

### B.1.2  MACRO-LEVEL ARCHITECTURE

**Macro-level design.** We only consider networks with exactly 4 stages. The first block of each stage (except Stage 1) reduces the resolution of both width and height by half, where the stride 2 is added to the first operator that is not conv1x1. Other blocks do not change the resolution. One block's output channel is either the same, or an integer multiple of the input channel.

**Macro-level encoding.** The 4 stages are divided apart by 3 '-' signs. For every stage, each block is represented by an integer, which shows the ratio between output and input channel for this block.

### B.1.3  ENCODING AS A WHOLE

Thus the whole backbone can be encoded by simply concatenating the block and macro encoding. The encoding of the whole network is formatted as:

$$\{Block\_ENCODING\}\_\{First\_CHANNEL\}\_\{Macro\_ENCODING\}$$

Some common architectures, including ResNet and Wide ResNet can be accurately represented by our encoding scheme, which is shown in Table B.1.3.

## B.2  NAS SEARCH ALGORITHM

### B.2.1  NON-DOMINATED SORTING ALGORITHM

Non-dominated sorting is mainly used to sort the solutions in population according to the *Pareto dominance principle*, which plays a very important role in the selection operation of many multi-objective evolutionary algorithms. In non-dominated sorting, an individual A is said to dominate another individual B, if and only if there is no objective of A worse than that objective of B and there is at least one objective of A better than that objective of B. Without loss of generality, we assume that the solutions of a population $S$ can be assigned to $K$ Pareto fronts $F_i$, $i = 1, 2, \ldots, K$. Non-dominated sorting first selects all the non-dominated solutions from population $S$ and assigns them to $F_1$ (the rank 1 front); it then selects all the non-dominated solutions from the remaining solutions and assigns them to $F_2$ (the rank 2 front); it repeats the above process until all individuals have been assigned to a *Pareto front*.

### B.2.2  NSGA-II: ELITIST NON-DOMINATED SORTING GENETIC ALGORITHM

To solve the problem in Eq. 1, Elitist Non-Dominated sorting genetic algorithm (NSGA-II) (Deb et al., 2000) is adopted to optimize the *Pareto front* $\mathcal{P}_f$ as shown in Algorithm 1. In this paper,

Table 8: The operations and channel changing ratios considered in our paper. Encoding for operators and ratios. $c$ stands for the channels of the current block.

| op number | operator(Input $c$) | ratio number | ratios |
|:---:|:---|:---:|:---:|
| 0 | conv3x3 | 0 | $\times 1/4$ |
| 1 | conv1x1 | 1 | $\times 1/2$ |
| 2 | conv3x3, group=2 | 2 | $\times 1$ |
| 3 | conv3x3, group=4 | 3 | $\times 2$ |
| 4 | conv3x3, group=$c$ | 4 | $\times 4$ |

Table 9: ResNets and Wide ResNets are represented by our encoding scheme. Basic Block is represented as '020-', as the two operators are both conv3x3 (denoted as '0'), and the output channel of the first operator equals to that of the block output (represented as '2'), and no other skip connection except the one connecting input and output; the macro-arch of ResNet 18 is encoded as '11-21-21-21', as each stage contains two blocks, where the first block in Stage 2, 3, 4 doubles the number of channels.

| model | encoding |
|---|---|
| ResNet18 | 020-_64_11-21-21-21 |
| ResNet34 | 020-_64_111-2111-211111-211 |
| ResNet50 | 10001-_64_411-2111-211111-211 |
| Wide ResNet50 | 11011-_64_411-2111-211111-211 |

we choose this kind of sample-based NAS algorithm instead of many popular parameter-sharing NAS method. This is because we want to further analysis of the sampled architectures and achieve insights and conclusions of the efficient training. The main idea of NSGA-II is to rank the sampled architectures by non-dominated sorting and preserve a group of elite architectures. Then a group of new architectures is sampled and trained by mutation of the current elite architectures on the $\mathcal{P}_f$ . The algorithm can be paralleled on multiple computation nodes and lift the $\mathcal{P}_f$ simultaneously. The mutation in the block-level search space includes adding new skip-connection, modifying the current operations and ratios. Meanwhile, the mutation in the macro-level search space includes randomly adding or deleting one block in one stage, exchanging the position of doubling channel block with its neighbor, and modifying the base channels. This well-known NSGA-II is easy to implement and we can easily monitor the improvement of each iteration. The stop criterion depends on the time limit or the computation cost constraints.

---

**Algorithm 1** Our modified NSGA-II Searching Algorithm
___

    **Input** Stop criterion, Search Space, number of computation nodes N.
1:  $t = 0$
2:  $P_t \leftarrow Random(A)$, generate a group of initial architectures.
3:  Evaluate $P_t$
4:  **while** not stop criterion **do**
5:      Apply non-dominated sorting to $P_t$ to obtain non-dominated fronts $A_i^*$
6:      Sort $A_i^*$ by Crowding distance and left top-N $A_j^*$ as Parents
7:      Create new generation $Q_t$ by mutation on current $A_j^*$
8:      Train the $Q_t$ on N computation nodes and Evaluate the accuracy of $Q_t$.
9:      $P_{t+1} \leftarrow Q_t \cup P_t$
10:    $t = t + 1$
11: **end while**
    **Output** The Pareto Optimal Front $A_i^*$

---

### B.3 NAS Implementation Details

#### B.3.1 Block-level search

In the phase of block-level search, a proxy task of ImageNet is created, which is a subset sampled fromits training set. This subset constitutes 100 labels, each of which has 500 images as the training set, and 100 as the validation set. We call this dataset ImageNet-100 in the following parts of this paper.

To avoid interference with macro architecture, the macro-level architecture is fixed to be the same as that of ResNet50. Each model is trained by ImageNet-100 with a batch size of 32 for 90 epochs and learning rate 0.1, which takes 3~10 hours on a single NVIDIA Tesla-V100 GPU. We do a random search at first, which uniformly samples all the valid blocks in the search space. Evolutionary Algorithm (EA) is then performed with three kinds of mutations: 1) replace one operator with another; 2) change the output channel of one layer; 3) Add/remove/modify a skip connection. We keep updating the Pareto Front between step time and accuracy during the whole process. As a

Table 10: The searched optimal efficient training models "ET-NAS" found by our NAS search. 'Acc' means the accuracy evaluated on the ImageNet; inference time and step time are measured in ms on single Nvidia V100, with a batch size of 64. By observing the optimal model, smaller models should use simpler blocks while bigger models prefer complex blocks.

| Model Name | Encoding | MParam | Gmac | MAct | Top-1 Acc | inf time (ms) | Training step time (ms) |
|---|---|---|---|---|---|---|---|
| ET-NAS-A | 2-_32_2-11-112-1121112 | 2.6 | 0.23 | 1.3 | 62.06 | 5.30 | 14.74 |
| ET-NAS-B | 031-_32_1-1-221-11121 | 3.9 | 0.39 | 1.3 | 66.92 | 5.92 | 15.78 |
| ET-NAS-C | 011-_32_2-211-2-111122 | 7.1 | 0.58 | 2.0 | 71.29 | 8.94 | 26.28 |
| ET-NAS-D | 031-_64_1-1-221-11121 | 15.2 | 1.55 | 2.6 | 74.46 | 14.54 | 36.30 |
| ET-NAS-E | 011-_64_21-211-121-11111121 | 21.4 | 2.61 | 4.7 | 76.87 | 25.34 | 61.95 |
| ET-NAS-F | 10001-_64_4-111-11122-1111111111111112 | 28.4 | 2.31 | 6.8 | 78.80 | 33.83 | 93.04 |
| ET-NAS-G | 211-_64_41-211-121-11111121 | 49.3 | 5.68 | 8.4 | 80.41 | 53.08 | 133.97 |
| ET-NAS-H | 10001-_64_4-111111111-211112111112-11111 | 44.0 | 5.33 | 10.9 | 80.92 | 76.80 | 193.40 |
| ET-NAS-I | 02031-a02_64_111-2111-21111111111111111111111-211 | 72.4 | 13.13 | 14.6 | 81.38 | 94.60 | 265.13 |
| ET-NAS-J | 211-_64_411-2111-21111111111111111111111-211 | 103.0 | 18.16 | 15.9 | 82.08 | 131.92 | 370.28 |
| ET-NAS-K | 02031-a02_64_1121-1111111111111111111111111111 -21111111211111-1 | 87.3 | 27.51 | 31.3 | 82.42 | 185.75 | 505.00 |
| ET-NAS-L | 23311-a02c12_64_211-2111 -21111111111111111111111-211 | 130.4 | 23.46 | 19.4 | **82.65** | 191.89 | 542.52 |

result, 10 blocks are selected as the candidates for the following rounds. Practically, during our search, the performance of early stop models aligns well with the fully-train accuracy. We checked the Spearman Rank Correlation for 103 architectures: $\rho = 96.6\%$. Thus, using early stop can greatly reduce the search cost by around 90% while keeping our NAS effective.

### B.3.2 MACRO-LEVEL SEARCH

We search the block-level architectures with the 10 blocks attained by block-level search. Random search is adopted at first, where the number of blocks is chosen randomly between 10 and 50, and the first and last channel is drawn from $\{32, 64, 128\}$ and $\{512, 1024, 2048\}$ respectively. EA search is then applied, where the mutations allowed are: 1) Add a '1'; 2) Remove a '1'; 3) Swap two different adjacent numbers. Similar to block-level search, Pareto Front between step time and accuracy is also kept updated.

Different from the previous phase, the whole ImageNet dataset is utilized for training. Each model is trained with a batch size of 1024 and a learning rate of 0.2 for 40 epochs.

### B.4 ET-NAS: MODEL ZOO INFORMATION AND THEIR ENCODINGS

After the NAS process done in Section B.3, 12 models are selected as our model

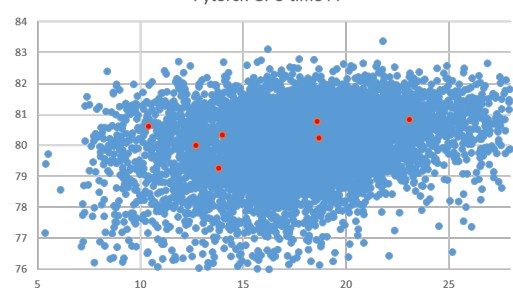

Figure 9: Results of the block-level search in ImageNet-100. The y-axis denotes the accuracy and x-axis denotes the latency. Blue dots are models searched in this step, while the red ones are Basic Block with first channel 64, 128, 192; Inverted Bottleneck Block (expansion rate 4) with first channel 64, 128; Bottle-Neck Block (expansion rate 4) with first channel 256, 320. It can be found that our algorithm can find more efficient block in the block-level search.

zoo ET-NAS of fine-tuning. Details of these models are shown in Table B.4. The inference time and step time are measured in ms on a single Nvidia V100, with batch size of 64. The resolution follows the standard setting of ImageNet: 224x224. By observing the optimal model in the table, smaller models should use simpler blocks while bigger models prefer complex blocks.

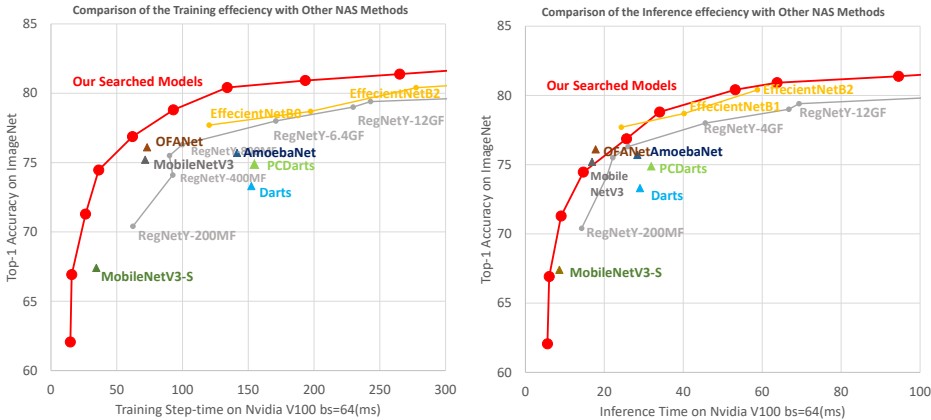

Figure 10: Comparison of the training and inference efficiency of our searched models (ET-NAS) with SOTA NAS models on ImageNet. We further compared our models with 8 other NAS results. It can be found that our method is more training efficient than some recent evolution-based NAS methods such as AmoebaNet (Real et al., 2019b), OFANet (Cai et al., 2019a) because of our effective search space.

Table 11: Regression Analysis: what makes a network efficient-training? We exam the effect of each component of network on the efficiency score. "Coef" and "SE Coef" are the estimated regression coefficient and standard error. "T-Value"/"P-Value" shows the significance of the variables.

| Block-level Regression Analysis | | | n=5500 | R-sq=56% | Macro-level Regression Analysis | | | n=1200 | R-sq=71% |
|---|---|---|---|---|---|---|---|---|---|
| **Terms** | **Coef** | **SE Coef** | **T-Value** | **P-Value** | **Terms** | **Coef** | **SE Coef** | **T-Value** | **P-Value** |
| *OP1 Channel Change Ratio* | -0.183 | 0.010 | -28.57 | 0.06 | ***Channel Size*** | **-0.210** | 0.023 | -9.26 | 0.00 |
| *OP2 Channel Change Ratio* | -0.168 | 0.010 | -27.75 | 0.02 | ***Double Channel Position 1*** | **-0.110** | 0.026 | -4.17 | 0.00 |
| ***Num of skip connection (add)*** | **-0.272** | 0.018 | -15.12 | 0.00 | *Double Channel Position 2* | 0.035 | 0.030 | 1.15 | 0.25 |
| ***Num of skip connection (concat)*** | **-0.362** | 0.018 | -19.93 | 0.00 | *Double Channel Position 3* | -0.016 | 0.030 | -0.54 | 0.59 |
| *Output_channel* | -0.539 | 0.010 | -53.57 | 0.00 | ***Double Channel Position 4*** | **0.224** | 0.025 | 9.14 | 0.00 |
| *conv3x3 (ref)* | 0.000 | - | - | - | *Double Channel Position 5* | 0.036 | 0.022 | 1.62 | 0.11 |
| *conv1x1* | -0.037 | 0.033 | -0.72 | 0.08 | ***Num block in Stage-1*** | **-0.562** | 0.023 | -24.69 | 0.00 |
| *conv3x3, w group=2* | 0.190 | 0.035 | 5.49 | 0.00 | ***Num block in Stage-2*** | **-0.139** | 0.023 | -6.04 | 0.00 |
| ***conv3x3, w group=4*** | **0.295** | 0.034 | 8.77 | 0.00 | ***Num block in Stage-3*** | **0.044** | 0.024 | 1.88 | 0.06 |
| ***Separable conv3x3*** | **-0.200** | 0.034 | -5.91 | 0.00 | *Num block in Stage-4* | -0.010 | 0.024 | -0.42 | 0.67 |

Figure 10 shows the comparison of our ET-NAS models with other SOTA ImageNet models. Inference time and training step time are measured in ms on a single Nvidia V100, with $bs = 64$. Our ET-NAS series show superior performance comparing to RegNet, EfficientNet series. Comparing to some EA-based NAS methods such as OFANet and Amoebanet, our method is also efficient in terms of training. We found that there exists a performance ranking gap between inference time and training step time in Figure 10. This is mainly due to the depth and the main type of operation of the models. We found that deeper networks with separable conv such as EfficientNet/MobileNet have a larger training-step-time/inference-time ratio comparing to our models (shallower&more common conv).

### B.4.1 WHAT MAKES A NETWORK EFFICIENT-TRAINING?

To answer this question, we first need to define a score for the efficiency of the searched models $\mathcal{A}$. In MOOP, the goodness of a solution is determined by *dominance*. Thus, we can use the non-dominated sorting algorithm to sort the $\mathcal{A}$ according to the Pareto dominance principle. Each architecture is assigned to one *Pareto front* and the rank $R_{\mathcal{P}}$ of that *Pareto front* can be regarded as the goodness of a solution, in our case, the efficiency. We then defined the *efficiency score* of $\mathcal{A}$ as: $s_E(\mathcal{A}) = -\frac{R_{\mathcal{P}}(\mathcal{A}) - \text{mean}(R_{\mathcal{P}}(\mathcal{A}))}{\text{std}(R_{\mathcal{P}}(\mathcal{A}))}$. Since *Pareto optimal front* is the Rank 1 *Pareto front*, larger efficiency score $s_E(\mathcal{A})$ means better efficiency.

Then we perform a multivariate linear regression analysis on the $\mathcal{A}_S$. According to our search space, ordinal/nominal variables that describe the model are denoted as predictors to fit the $s_E(\mathcal{A})$. Table 11 shows the coefficients from the regression analysis on both block-level and macro-level

designs. Positive coefficients indicate a positive relationship. "P-Value" shows the significance of the variables. We summarize and highlight several noteworthy conclusions uncovered by our analysis:

- By observing optimal $\mathcal{A}_i^*$, smaller models should use simpler blocks while bigger models prefer complex blocks. Simply increasing depth/width to expand the model in Tan & Le (2019) may not be optimal.
- Adding additional skip connections will decrease the training efficiency of the model (The Coef is significantly negative). Using "add" to combine the features is more efficient than "concat".
- Using "conv3x3, w group=4" is the best operation among the searched operations (Coef is 0.295). Separable conv3x3 is not efficient for training (Coef is -0.2).
- The first double-channel position should be more close to the beginning of the network, while the final double channel-position should be delayed to the end of the network.
- Fewer blocks should be assigned to the first two stages. More should be assigned to the 3rd stage.

### B.5 $CO_2$ CONSUMPTION ANALYSIS

Fine-tuning from the pretrained ImageNet/language model is a de-facto practice in the deep learning field (CV/NLP). Our NASOA improves the efficiency of fine-tuning which has the potentials to greatly reduce computational cost in GPU clusters/cloud computing. According to a recent study (Strubell et al., 2019b), developing and tuning for one typical R&D project (Strubell et al., 2018) in Google Cloud computing needs about \$250k cost, 82k kWh electricity, and 123k lbs $CO_2$ emission, which equals to the $CO_2$ consumption of air traveling (NY$\leftrightarrow$SF) 62 times. Among most of them, 123 hyperparameter grid searches were performed for new datasets, resulting in 4789 jobs in total. It is believed that the proposed faster fine-tuning pipeline can save up to 40x computational cost among them. Furthermore, we have released all the searched efficient models to help the public skipping the computation-heavy NAS stage and directly enjoy the benefit of our methods. In conclusion, our NASOA is meaningful for environment protection and energy saving.

### B.6 DETAILED ALGORITHMS OF NASOA

Detailed algorithms of our Model Zoo (ET-NAS) search can be found in Algorithm 2. The pseudo code of online fine-tuning schedule generator training, prediction, and update can be found in Algorithm 3.

---

**Algorithm 2** Efficient Training Model Zoo (ET-NAS) Creation

**Input:** Block/Macro Search Space $\mathcal{S}_i, \mathcal{S}_a$, Stop Criterion $\Gamma$, #Computation Nodes $K$, Sensitive Factor $\epsilon$, #Block Architectures $M$, #Models in Model Zoo $N$.
**Output:** Final Model Zoo $Z_{oo}$
1: **procedure** BLOCKSEARCH($\mathcal{S}_i, \Gamma, K, \epsilon, M$)
2:     $P_f \leftarrow$ NSGA-II($\Gamma, \mathcal{S}_i, K, \epsilon$)       ▷ Our modified NSGA-II, see Algorithm 1
3:     $Cells \leftarrow$ MOSTCOMMON($P_f, M$)        ▷ Most common $M$ cells from $P_f$
4: **end procedure**
5: **procedure** MACROSEARCH($\mathcal{S}_a, \Gamma, K, \epsilon, N$)
6:     $P_f \leftarrow$ NSGA-II($\Gamma, \mathcal{S}_a(Cells), K, \epsilon$)
7:     $Z_{oo} \leftarrow$ NSGASORT($P_f, N, \epsilon$)       ▷ Choose models based on crowding-distance
8: **end procedure**

---

## C IMPLEMENT DETAILS OF HPO METHODS

We use BOHB and random search in our experiments as the HPO baseline. As stated in Section 4.4, The shortest/longest time constraint (budget) is defined as the time of fine-tuning 10/50 epochs for ResNet18/ResNet101 and the rest are equally divided into the log-space, which can be represented as: $t_x = t_0 * (t_3/t_0)^{x/t_3}$, where $x = [0, 1, 2, 3]$, $t_0$ is the time to train ResNet18 for 10 epochs and $t_3$ is the time to train ResNet101 for 50 epochs.

We only compare the HPO setting under the same max computational budgets equal to $t_1$ in Table 5 (left). For random search, we randomly sample candidates from predefined search space until

---

**Algorithm 3** Online Fine-Tuning schedule Generator Training, Prediction, and Update

---

**Input:** Model Zoo $Z_{oo}$, Time Evaluator $T_s$, Acc Evaluator $T_r$, Hyper-parameter Space $\mathcal{S}_{HP}$, Known Datasets $D_{old}$, New Dataset $D_{new}$, #Meta-data $H_M$, Time Constraint $T_l$, #Configurations $H$.

**Output:** Optimal Model $\mathcal{A}^*$, Hyper-parameters $Regime^*_{FT}$, Predictor $Acc_P$

1: **procedure** OFFLINETRAINING($Z_{oo}, T_r, \mathcal{S}_{HP}, D_{old}, H_M$)
2:     $MetaData \leftarrow \emptyset, Acc_P \leftarrow$ ADAPTIVEMLP(.)          ▷ Initialize default predictor
3:     **for** $D \in D_{old}, i \leftarrow 1$ to $H_M$ **do**
4:         $\mathcal{A}, Regime_{FT} \leftarrow$ RANDOM($Z_{oo}, S_{HP}$)      ▷ Randomly select from search space
5:         $Acc \leftarrow T_r(D, \mathcal{A}, Regime_{FT})$          ▷ Train with selected configuration
6:         $MetaData \leftarrow MetaData \cup \{(\mathcal{A}, Regime_{FT}, Acc)\}$     ▷ Add this result to meta-data
7:     **end for**
8:     $Acc_P \leftarrow$ TRAIN($Acc_P, MetaData$)          ▷ Train predictor with all meta-data
9: **end procedure**
10: **procedure** ONLINEPREDICTION($Z_{oo}, D_{new}, Acc_P, T_l, T_s, H$)
11:     $MetaData \leftarrow \emptyset$
12:     **for** $i \leftarrow 1$ to $H$ **do**
13:         $\mathcal{A} \leftarrow$ RANDOM($Z_{oo}$)
14:         $Epoch \leftarrow T_l \div T_s(\mathcal{A}, D_{new})$          ▷ Always choose the largest epoch within $T_l$
15:         $Regime_{FT} \leftarrow$ RANDOM($S_{HP}|Epoch$)       ▷ Randomly select condition on $Epoch$
16:         $MetaData \leftarrow MetaData \cup \{(\mathcal{A}, Regime_{FT})\}$
17:     **end for**
18:     $\mathcal{A}^*, Regime^*_{FT} \leftarrow$ PREDICT($G, MetaData$)          ▷ Choose the optimal from $H$ configs
19:     $Acc \leftarrow T_r(D_{new}, \mathcal{A}^*, Regime^*_{FT})$
20:     $Acc_P \leftarrow$ TRAIN($Acc_P, \{\mathcal{A}^*, Regime^*_{FT}, Acc\}$)     ▷ Improve predictor with this meta-data
21: **end procedure**

---

reaching the max computational budget. And for BOHB, we use the opensource implementation of BOHB at `https://github.com/automl/HpBandSter`. We set random fraction=0.3, percent of good observations=15%, min budget=25% and max budget=100% with respect to our max computational budget. And we use BOHB with 40x computational cost than our proposed methods with different model zoos. The results are presented in Section 4.4.

