# OpenReview forum: "NASOA: Towards Faster Task-oriented Online Fine-tuning"
_ICLR.cc/2021/Conference — Reject_

### Official Review · AnonReviewer2 · 2020-10-27
**This paper proposes a joint Neural Architecture Search and Online Adaption (NASOA) framework to achieve a faster task-oriented fine-tuning upon the request of users. My biggest question is why the fine-tuning should be based on a specific network.**

**Rating:** 7
**Confidence:** 3

**Review:**

In this paper, a joint Neural Architecture Search and Online Adaption (NASOA) framework is proposed to achieve a faster task-oriented fine-tuning upon the request of users. In particular, two main contributions are made in this paper: (1) A fine-tuning pipeline that seamlessly combines the training-efficient NAS and online adaption algorithm is introduced, which can effectively generate a personalized fine-tuning schedule of each desired task via an adaptive model for accumulating experience from the past tasks. (2) A block-level and macro-level search space is introduced in the resulting framework, which enables a simple to complex block design and fine adjustment of the computation allocation on each stage.

I have two  comments about this paper， which  go as follows:

(1)	In this work, a joint NSAOA framework is introduced facilitate a fast continuous cross-task model adaption. Why not learn an effective fine-tuning regime based on a hand-crafted network? In my point of view, a good fine-tuning regime should be also independent of the choice of network.

(2)	The block-level and macro-level search space can reduce the redundant skip-connections in the resulting structures. To the best of my knowledge, the skip-connection is effective to avoid the gradient vanishing problem in a very deep neural network, therefore the skip-connections at the lower layers are very important in the network design. However, the skip-connections at the lower layers will take more memory cost at both training and testing stages. From this point of view, authors should explain where the redundant skip-connections always occur in the rebuttal.

---

> ### Author Response · Authors · 2020-11-12
> **Thank you so much! Here are our responses to your comments.**
>
> Thank you so much for your review. We are happy to discuss our paper with you.  We will keep polishing our paper based on your opinion during the rebuttal period.
>
> **Q: Why the fine-tuning should be based on a specific network？Why not learn an effective fine-tuning regime based on a hand-crafted network?**
>
> **A:** Our goal is to provide the best fine-tuning regime according to the user’s demand, such as the constraint on maximum training time. Different network structures have different convergence speed and perform differently under various conditions. Therefore, we regard the fine-tuning hyperparameters settings and network architectures as a whole. We aim to find the best combination under a specific constraint rather than only finding the best fine-tuning regime for a certain network.
>
> Furthermore, our fine-tuning regime generator can be easily applied to any existing fixed hand-crafted network structures. For example, in Table 4, we compare our fine-tuning regime generator and other HPO to generate fine-tuning regimes only on a specific network (RegNetY-16F in Table 4). Table 5 further shows the effectiveness of our regime generator: Comparing to using "BOHB(HPO only)", "Our OA only” can increase the fine-tuning accuracy by 6.54% and reduce the computational cost by 40X.
>
> **Q: Pros and cons of skip-connections. Discussion on skip-connections.**
>
> **A:** As pointed out by the reviewer, skip-connections effectively avoid gradient vanishing problem, but will take more memory and increase memory access cost (MAC). We agree with you.
>
> For our searched cell architecture (examples and encodings can be found in Figure 7 and Table 10 in appendix), it can be found that our searched cells have limited skip-connection (usually 1, at most 3) comparing to some existing NAS methods such as DARTS, ENAS (more than 10). Thus, our search space can keep the training both effective (avoid the gradient vanishing problem) and efficient (less memory access cost).
>
> Furthermore, the side effects on memory can greatly be reduced by avoiding long skip connections since the memory can immediately be released after the skip connection. That is why our search space only has skip connections within blocks.
>
> **If you have further questions or anything unclear, we are more than happy to discuss them during the rebuttal period. Thanks again for your kind review.**

---

### Official Review · AnonReviewer3 · 2020-10-28
**Initial review from R3**

**Rating:** 7
**Confidence:** 4

**Review:**

This paper is the first to address the problem of searching better backbone architectures for downstream tasks and online hyper-parameter selection.  The whole system proposed in this algorithms offers an end-to-end solution for producing a well-trained architecture for a specific downstream task within a fixed training budget. I believe this system will have a large impact in industry model deployment.
This paper separate the overall searching process into (1) generating efficient training model zoo (pretrained on a source dataset) (2) task-oriented fine-tuning schedule (Mode selection, hyper-param selection for downstream task fine-tuning).  The search space of the efficient model zoo is decoupled to two stages: block-level search space and macro-level search space. This design could enrich the model zoo with diverse models architectures.   The major part of task-oriented fine-tuning schedule is a performance predictor which is fed with target dataset embedding, model identity and training hyper-params and output an assessment of the final performance.

Through extensive evaluation experiments, the effectiveness is of this system is demonstrated.  Through well defined ablation studies, the author offers our insightful conclusion of why their systems works:
1. A better and more diverse model zoo, where  smaller models have simper block structures and larger models have more complicated block structure.
2. The well-trained performance predictor in task-oriented fine-tuning  scheduler can successfully capture some sort of correlation between bunch of hyper-param and final performance.

Some of my concerns:
1. The searched architectures are not  directly optimized for down-stream task performance.
2. This system cannot address the problem with consistently changing down-stream task, and the cost of training the scheduler for a new downstream task is a little bit huge because of the large pretraining model zoo.

Overall, I recommend this paper to be accepted due to its engineering efforts, several interesting conclusion drawn from empirical study and great impact in industries.

---

> ### Author Response · Authors · 2020-11-12
> **Thank you so much! Here is your reply.**
>
> We are glad that you think our work is interesting. All your comments are summarized and addressed as follows.
>
> **Q: The searched architectures are not directly optimized for down-stream task performance.**
>
> **A:** There are two reasons that our NAS is conduct on the ImageNet rather than down-stream tasks:
>
> a) In the practical setting of fine-tuning on the cloud computing, the upcoming down-stream tasks are not available. It is impossible to directly optimize our architectures for down-stream task performance.
>
> b) Suggested by [1], the model fine-tuning accuracy (model generalization ability) has a strong correlation between ImageNet accuracy (r = 0.96). Thus, we can boost the performance of  down-stream tasks by optimizing our architectures on ImageNet.
>
> **Q: This system cannot address the problem with consistently changing down-stream task.**
>
> **A:** Our NASOA can easily handle upcoming, consistently changing down-stream tasks due to our online fine-tuning regime generator. Our regime generator can automatically select the most suitable model and fine-tuning schedule for the user-specific task. Note that our online adaption behaves in a one-shot fashion and doesn’t involve additional searching cost as HPO, endowing the capability of quickly providing various training regimes for consistently changing down-stream tasks. Furthermore,  benefiting from our online learning algorithm, the diversity of the data and the increasing results can further continuously improve our regime generator. Figure 4 show that our methods can improve around 2.1%˜7.4% accuracy on average for different kinds of tasks (including Fine-Grained, Medical Img, Scene cls tasks).
>
> **Q: The cost of training the scheduler for a new downstream task is a little bit huge because of the large pretraining model zoo.**
>
> **A:** After a new downstream task comes, we only need to fine-tune our regime predictor by the new observation. It only requires several steps of hedge backpropagation updates on a very small MLP with 64 channel size. The finetuning can even be conducted on a CPU (don't need GPU). The training complexity is fixed and won't grow big when we increase the models in the model zoo.
>
> **If you have further questions or anything unclear, we are more than happy to discuss them during the rebuttal period. Thanks again for your kind review.**
>
> [1] Simon Kornblith, Jonathon Shlens, and Quoc V Le. Do better imagenet models transfer better? In CVPR, 2019.

---

### Official Review · AnonReviewer4 · 2020-10-28
**Official Blind Review #4 - post rebuttal update**

**Rating:** 6
**Confidence:** 3

**Review:**

SUMMARY:

The paper presents a joint Neural Architecture Search (NAS) and Online Adaption (OA) framework named NASOA, aiming at providing faster task-oriented ﬁne-tuning system. The first step of the approach is an offline NAS to form a pretrained training-efficient model zoo, which is followed by an online scheduler which selects the most suitable model and generates a personalized training regime with respect to the target task.
***************************************************************************
STRENGTHS
- The paper is fairly well written and easy to follow.
- The proposed approach is fairly novel: up to my knowledge, this is the first effort to combine NAS and Online Adaptation techniques for faster fine-tuning task. In addition, the proposed joint/block macro level search space is novel too.
- The practical benefits of the proposed approach are obvious for AutoML systems, and the shared ET-NAS model zoo will help other researchers working on this topic.
- The paper provides (in appendix B) a short but interesting analysis on CO2 consumption. This is a good practice which is rare enough to be worthy of note.
***************************************************************************
WEAKNESSES AND REMARKS
- I have some concerns regarding the fact that the paper does not compare the proposed approach to a baseline offline setting. The idea would be to collect offline training data by finetuning on several datasets (of different nature) and collect the corresponding accuracies to train a predictor (e.g. an MLP). The paper mentions this possibility, but says that it is less realistic than the online setting. Which is true, but it would have been interesting to compare the two settings
- I’m not sure to understand the relevance of the preliminary experiments presented in sub-section 4.1. The findings (i.e. 1. fine-tuning performs better than training from scratch, and 2. learning rate and frozen stage are crucial for ﬁne-tuning) seem obvious to me, and correspond to widely-adopted common practices.
- It’s not clear to me how the models are organized into groups in Table 2. Are they grouped based on their respective complexity? If so, it should explicitly mentioned by adding a column with the number of parameters of each model.
- When presenting the ImageNet results in Table 2, the paper does not mention that current sota on ImageNet is around 88% top-1 accuracy, i.e. about 6 points above the best result presented in the Table. The paper should add this information to slightly tone down the claim that ‘searched models outperform other SOTA ImageNet models in terms of training accuracy’.
- It would have been interesting to give more details about the NSGA-II algorithm in the paper itself, rather than only referring to the appendix. I understand that it’s due to the lack of space, but it is a little bit frustrating to have no information in the paper about this point, especially about the modifications compared to the original algorithm.
- Figures 3 and 4 are quite useless in their current form, since the text size is too small to be read.
- Table 4 is hardly interpretable without reading its corresponding descriptive paragraph in the paper. Authors should give more information (e.g. in the caption) about each method.
- Page 5, Line 5: MLP stands for Multi Layer Perceptron (not Perception).
***************************************************************************
JUSTIFICATION OF RATING

Despite the concerns mentioned above, I think the proposed approach is an interesting addition to the task-oriented online fine-tuning literature, especially considering its practical benefits for AutoML systems. I also think the paper could be considerably improved by taking into account the comments made above. Overall, I’m leaning to reject (5: Marginally below acceptance threshold), but I would be considering to increase my rating if some modifications are made during rebuttal.

***************************************************************************
POST REBUTTAL UPDATE

The authors provided a detailed rebuttal which addressed many of my concerns, and clarified some points. I will update my rating from 5 (marginally below acceptance threshold) to 6 (marginally above acceptance threshold).

---

> ### Author Response · Authors · 2020-11-12
> **Thank you for your  constructive and insightful  review! Here are our responses to your comments.**
>
> Thank you so much for your constructive and insightful comments!! We are glad that you think this is a nice paper.
>
> We are polishing the paper based on your valuable suggestion, e.g., enlarging Figures 3 and 4, adding more explanation in the caption of Table 4, adding additional experiments for the baseline of offline setting and changing Multi-Layer Perception to Perceptron. We will try to move some key aspects in supplementary back to the main document, for example, some explanation of NSGA-II.
>
> Your comments are addressed as follows:
>
> **Q: To compare the proposed approach to a baseline offline setting.**
>
> **A:** As you suggested, we are adding an experiment to compare the offline setting (fixed MLP) on the performance of the downstream tasks with our NASOA. The results will be updated in a few days. Thank you so much for pointing it out. Besides, Table 3 shows our online adaptive methods can achieve better prediction performance than the offline setting (fixed MLP). Our method enjoys the benefit from the adaptive depth, which allows faster convergence in the initial stage and strong predictive power in the later stage.
>
> **Q: Some of the Findings in sub-section 4.1 preliminary experiments seem obvious.**
>
> **A:** Some of the findings are indeed obvious for some experts. We put them here just in case some readers who are not familiar with this field may feel a little bit gap without this sub-section. (That's why we put most  preliminary experiment details in the appendix.) From this section, we also draw some important conclusions related to the motivation of our paper, such as  a) For an efficient fine-tuning, the model matters most. b) We should select the suitable model according to the task and time constraints. We will further remove some obvious conclusions in the main text.
>
> **Q: How are the models in Table 2 grouped?**
>
> **A:** They are grouped by Top-1 Accuracy of ImageNet. (Top-1 Acc, 70~72, 73~75, 75~77, 77~79,79~81, 81+) We aim to compare the inference time, training step time and accuracy in this table. The number of parameters is not shown here. They can still be found in Table 10 in Appendix.
>
> **Q: ET-NAS models are not SOTA on ImageNet, since the highest accuracy is around 86% (without extra data).**
>
> **A:**  When referring to SOTA in this paper, we mean the trade-off between training accuracy and training speed. In other word, ET-NAS models achieve the highest accuracy among models whose training step time are no larger than them. Indeed, our highest accuracy does not reach the highest one in the industry. we will tone down our claim by adding the constraint of training speed.
>
> By the way, the "noisy student" method reaches the highest ImageNet accuracy of 88%. They used additional 3.5B Instagram Images to train the model. The EfficientNet-B7 using only ImageNet data reaches 84.4%. It requires several days of training with hundreds of TPUs. In contrast, our biggest model (ET-NAS-L: 82.7%) is training efficient and can be trained on an 8-GPU machine within 4 days.
>
> **Q:  Details of modified NSGA-II, especially the differences from the original one.**
>
> **A:**  We are adding more details about the NSGA-II algorithm in the main paper. Details of modified NSGA-II can be found in Algorithm 1 in Appendix. We modify the NSGA-II algorithm to become a NAS algorithm: a) To enable parallel searching on `N` computational nodes, we modify the `non-dominated-sort ` method to generate exactly `N` mutated models for each generation, instead of a variable size as the original NSGA-II does. b) We define a group of mutation operations for our block/macro search space for NSGA-II to change the network structure dynamically. c) We add a parent computation node to measure the selected architecture's training speed and generate the Pareto optimal models.
>
> **Q:   Authors should give more information (e.g. in the caption) about each method in Table 4.**
>
> **A:**  Thank you for pointing it out. We are adding more description in the caption of Table 4.
>
> **We will update our paper as soon as possible. If you have further questions or anything unclear, we are more than happy to discuss them during the rebuttal period. Thanks again for your kind review.**

---

> > ### Author Response · Authors · 2020-11-13
> > **We have uploaded a revision of our paper.**
> >
> > Hi, we have uploaded a revision of our paper according to your suggestions. The change log can be found in the first comment. Thank you so much! If you have further questions or anything unclear, we are more than happy to discuss and revise them.

---

### Official Review · AnonReviewer1 · 2020-10-28
**A practical fine-tuning method but lacks interesting novelty.**

**Rating:** 3
**Confidence:** 3

**Review:**

This works aims at task-oriented fine-tuning from pre-trained ImageNet models. It proposes a Neural Architecture Search and Online Adaption framework (NASOA) to perform fast task-oriented model fine-tuning. The NASOA first employ an offline NAS to select a group of models and then pick up the most suitable model from this group via an online schedule generator.

Weaknesses:
The proposed method may be useful in some fine-tuning scenarios, however will have low overall impact. From the reviewer's view, lack of novel and interesting part. The proposed approach simply uses and combines existing methods, e.g., NAS, seems more like an engineering project.

---

> ### Author Response · Authors · 2020-11-12
> **We humbly request  a more serious and technical review for our paper.**
>
> We are more than disappointed to receive such a review **without any technical remarks**. In contrast to your opinion, even the other reviewers think our work is novel and will significantly contribute to the community.
>
> As mentioned by Review #4,
>
> - “**The proposed approach is fairly novel**: up to my knowledge, this is **the first effort** to combine NAS and Online Adaptation techniques for faster fine-tuning task. In addition, the proposed joint/block macro level search space **is novel too**.”
> - “The **practical benefits** of the proposed approach **are obvious for AutoML system**s, and the shared ET-NAS model zoo **will help other researchers working on this topic**.”
>
> Also, quote by Review #3,
>
> - “Overall, I recommend this paper to be accepted due to its engineering efforts, **several interesting conclusion** drawn from empirical study and **great impact in industries**.”
>
> We want to **emphasize our novelty and contribution again**:
>
> 1. We are the first to propose a faster fine-tuning pipeline that combines the training-efficient NAS and online adaption algorithm. It is the first time in the literature to consider the effectiveness of the model architectures for the faster fine-tuning problem.
> 2. The proposed novel joint block/macro level search space enables a flexible and efficient search. The resulting model zoo ET-NAS is more training efficient than very strong ImageNet models. (All the pretraining models have been released.)
> 3. The whole NASOA pipeline achieves much better fine-tuning results in terms of accuracy and fine-tuning efficiency than current fine-tuning best practice and HPO method, e.g. 40x faster than BOHB method.
>
> We humbly request  a more serious and technical review of our paper from you. We are more than happy to receive your solid review and polish our paper during the rebuttal period. Hope to hear from you soon.

---

### Author Response · Authors · 2020-11-13
**Rebuttal Revision of our paper**

**Update Log of our paper**

- Add a new experiment of the offline baseline "Our Zoo with Fixed MLP Predictor (Offline)" in Table 4. Also, update Section 4.4 with the new results.
- Put the comparison of our searched models (ET-NAS) with 8 other NAS results in Figure 4. Enlarge Figure 3 and the font of Figure 5.
- Add more explanations of each method in the caption of Table 4.
- Add some explanation of NSGA-II in Section 3.1 Multi-objective Searching Algorithm.
- Refine Section 4.1 Preliminary Experiments with less obvious results.
- Fix some typos such as Multi-Layer Perception to Perceptron.

---

> ### Author Response · Authors · 2020-11-18
> **If you have any comments or concerns, we are willing to discuss and polish our paper.**
>
> If you have any comments or concerns, we are willing to discuss and polish our paper.

---

### Decision · Program_Chairs · 2021-01-07
**Final Decision**

**Decision:**

Reject

**Comment:**

In this paper, a network architecture search (NAS) problem in a changing environment is studied and an online adaptation (OA) algorithm for the problem is proposed. Many reviewers found that the OA-NAS problem discussed in this paper is interesting and practically important. However, many reviewers (including those with high review scores) recognize that the weakness of this paper is the lack of sufficient theoretical verification. Furthermore, although extensive experiments are conducted, it is still not clear whether the experimental setups discussed in the paper are generally applicable to other practical problems. Overall, although this is a nice work in that a new practical problem is considered and a workable algorithm for the problem is demonstrated in an extensive simulation study, I could not recommend the acceptance in its current form because of the lack of theoretical validity and evidence of general applicability.